# A qualitative exploration of clinicians' strategies to communicate risks to patients in the complex reality of clinical practice

**Romy Richter** [1]*, **Esther Giroldi**[1,2], **Jesse Jansen**[1], **Trudy van der Weijden**[1]

**1** Department of Family Medicine, School Care and Public Health Research Institute (CAPHRI), Faculty of Health Medicine and Life Sciences (FHML), Maastricht University, Maastricht, Limburg, Netherlands, **2** Department of Educational Development and Research, School of Health Professions Education (SHE), Faculty of Health Medicine and Life Sciences (FHML), Maastricht University, Maastricht, Limburg, Netherlands

* romy.richter@maastrichtuniversity.nl

## Abstract

### Background

Risk communication, situated in the model of shared decision making (SDM), is an essential element in daily clinical practice. The scientific literature makes a number of generic recommendations. Yet the application of risk communication remains a challenge in patient-clinician encounters. How clinicians actually communicate risk during consultations is not well understood. We aimed to explore the risk communication strategies used by clinicians and extract narratives and visualizations of those strategies to help inform medical education.

### Methods

In this qualitative descriptive study, we interviewed fifteen purposely sampled clinicians from several medical disciplines, who were familiar with the concept of SDM. Deductive and inductive content analysis was used during an iterative data collection and analyses process.

### Results

Our study identified various strategies reported to be used by clinicians to address the complexities of risk communication such as dealing with uncertainty. These included verbal, numerical and visual risk communication and framing. Clinicians were familiar with recommended risk formats such as natural frequencies and population pictograms. However, it became clear that clinicians' expertise and communication goals also play an important role in the risk talk. Clinicians try to lay a foundation for balanced decision-making and to incorporate patient preferences while faced with several challenges such as the dilemma of raising awareness but triggering anxiety or fan fear in patients. Consequently, they also use communication goals such as influencing mindset and reassuring patients. Additionally, clinicians frequently have to account for the illusion of certainty in the risk talk.

**Data Availability Statement:** All relevant data are within the manuscript and its supporting Information files. Due to ethical restrictions on sharing data which contain potentially personally

identifiable information, the transcribed interviews are available on request.

**Funding:** The author(s) received no specific funding for this work.

**Competing interests:** The authors have declared that no competing interests exist.

## Conclusion

Risk communication is a multi-faceted construct that cannot be dealt with in isolation from the clinical context. For future research we recommend considering a more practical framework within the clinical setting and to take a goal-directed approach into account when investigating and teaching the topic. The patient perspective should also be addressed in further research.

## Background

Modern healthcare faces the shift from the paternalistic model of doctor-patient communication ("doctors know best") to a model of shared decision making (SDM) based on sharing information and incorporating patient preferences and values into decision-making [1–4]. The latter concept is based on an iterative approach comprising three phases: team talk, option talk and decision talk. The communication of risks and benefits (risk communication) is an essential part of the option talk in which various screening, diagnostic, treatment or palliative alternatives are compared [2,4–9].

In recent years, the focus has increasingly been on the varying formats and methods of communicating risks to patients in an effective way [10–12]. It has been established that certain risk formats affect the perception of risk and subsequent decisions [13–18]. Broadly speaking, three formats of communicating risk can be distinguished: verbal, numerical and graphical. Verbal risk communication refers to the use of descriptive labels such as "high risk" or "low risk". This qualitative description of risks leaves room for interpretation and thus leads to ambiguity of definition [19,20]. Consequently, a growing body of scientific evidence points to the communication of risk in numerical terms. Numerical formats include percentages, e.g. 20% probability of a heart attack, and natural frequencies, e.g. 20 in 100 cases. Natural frequencies seem to be easier to understand than percentages [6,10,21,22]. Notwithstanding the growing supportive body of evidence for numeric risk communication formats, risk communication faces challenges. Numerical health illiteracy has been shown in patients, and also in many clinicians [10,12,21,23,24]. In recent years, a number of researchers have addressed the graphical display of risks by methods such as population pictographs (icon arrays), bar charts and risk ladders [5,25–28]. While pictograms have been proven useful in clinical conversations [20,29,30], however, overall the understanding of graphical risk presentation remains unclear [31,32].

Risk information can be given in a number of ways; this is known as framing [12,33]. Risk messages can be framed in terms of gain versus loss (goal framing). For instance, taking a medication will increase the chance of not getting a disease versus not taking the medication will increase the chance of getting the disease. Another approach is to describe the information in terms of positive versus negative framing (attribute framing), e.g. a patient can either be told there is an 80% chance of survival or a 20% chance of dying [12,33,34]. Choice of framing influences the perception of risks and hence the decision made during a clinical consultation [15].

An inherent part of risk communication is uncertainty, which can take multiple forms. A number of frameworks have classified uncertainty [35,36], however, there is still no consensus on a uniform categorization of uncertainty [12,37,38]. Looking at the judgement and decision-making literature, two dimensions of uncertainty are often distinguished: aleatoric and epistemic. Aleatoric uncertainty refers to the natural randomness in a process (stochastic uncertainty). Each time an experiment is conducted under similar conditions, the outcome may

differ. Epistemic uncertainty results from limited data and knowledge of a fact (that is either true or not true) [39]. Humans in general, thus patients and doctors, are faced with the illusion of certainty in daily and clinical life. A clear presentation of uncertainty seems to be one of the most difficult parts of communicating risks [12,37]. Presentation formats can vary from verbal to numerical or visual. The effect of communicating uncertainty has not been well studied, hence it is unclear what types of uncertainty should be communicated and to what extent [38,40]. Efficient ways to communicate uncertainty are still under debate and there is little guidance on best practice approaches [12,37,38].

Despite theoretical insights into recommendations for preferred risk communication formats [10,12,21,34,41,42], risk communication remains a major challenge [9,12,21,23]. The various medical disciplines with their different contexts demand clinicians to nuance the risk talk sensitively to specific aspects of their discipline. For example, in clinical genetics risk communication needs to address predictive testing such as the small chance of getting a false positive test result or the small chance of miscarriage in invasive testing [43]. Thus, clinical geneticists often have to deal with communicating small and difficult to imagine risks that could have a great impact in the future of the patient. General practitioners often counsel the patient for more general problems in the present, while doctor and patient usually have a closer and continuous relationship with repetitive opportunities for a dialogue and reassurance of the patient. Whereas surgeons often have to communicate the side effects and consequences of severe surgical procedures under time pressure while having seen the patient one or two times in the hospital. On the other hand, oncologists often have to consider the communication of overtreatment and overdiagnosis when discussing screening and treatment options, especially when dealing with frail elderly patients. Undoubtedly, risk communication is a core skill for clinical counsellors in various medical disciplines. However, as far as we know, the literature mainly provides formalistic or standardized language on written communication of numerical and visual risk communication formats and lacks practical and illustrative examples of how to communicate risks to patients in daily clinical practice. Research emphasizes that trainees are in need of concrete illustrations of meaningful language and graphical examples in order to acquire complex communication skills [44,45]. The provision of illustrative risk communication strategies in the SDM concept could support educational training programs. Altogether, this points to a potential gap between theoretical recommendations in literature and ease of application in clinical practice.

The aim is to add to the existing literature on risk communication by gaining a deeper understanding of the clinician's actual strategies in communicating risk in daily clinical practice. A comprehensive insight into the best practice approaches of a sample of clinicians who have experience of, or are at least familiar with, the concept of SDM, may provide valuable examples of real life risk communication strategies, which could support trainees in the process of acquiring SDM skills [44]. In line with Lingard´s theory on communities of expertise, we took the view that as members of a community, clinicians have developed a certain professional expertise on how to communicate risk to patients [46,47]. To promote the development of risk communication training for trainees, we explored the strategies used by clinicians to communicate risk to patients and aimed to extract illustrative examples (narratives) and visualizations of these strategies.

## Methods

### Study design

We conducted a qualitative descriptive study with semi-structured interviews. We took the ontological orientation of relativism, which holds the view that multiple subjective realities

exist. Consequently, we chose the epistemological orientation of subjectivism referring to the existence of many interpretations of reality, and to clinicians' risk communication strategies being subjectively developed in relation to the clinical situation [48]. The consolidated criteria for reporting qualitative research (COREQ) has been used to guide reporting of the research [49].

## Ethical approval and informed consent

The NVMO Ethical Review Board granted approval for our study. Clinicians were informed about the study via e-mail. In this e-mail the research was described, and the goal was stated, and the clinicians were asked whether they like to participate in a qualitative interview. Subsequently, clinicians gave written informed consent (via e-mail) whether they were willing to participate. Further, verbal consent was obtained before the start of the interview, just before the audiotape had been started. Verbal informed consent was given for audiotaping and using the data for publication in a scientific journal. Verbatim transcripts of the recorded interviews were anonymized with codes.

## Conceptual framework

Risk communication is defined as: "The open, two-way exchange of information and opinion about risk, leading to a better understanding of the risk in question, and promoting better (clinical) decisions about management" [34,50]. We sited risk communication as the so-called "risk talk" within the option talk in accordance with one of the models of SDM [2,9]. In order to display a theoretical foundation and establish the current state of knowledge regarding risk communication strategies [10,12,34,41] a deductive conceptual framework based on a literature review was generated (Fig 1). This framework steadily guided the data collection and analysis process. Other risk communication strategies, clinician goals, content and contextual

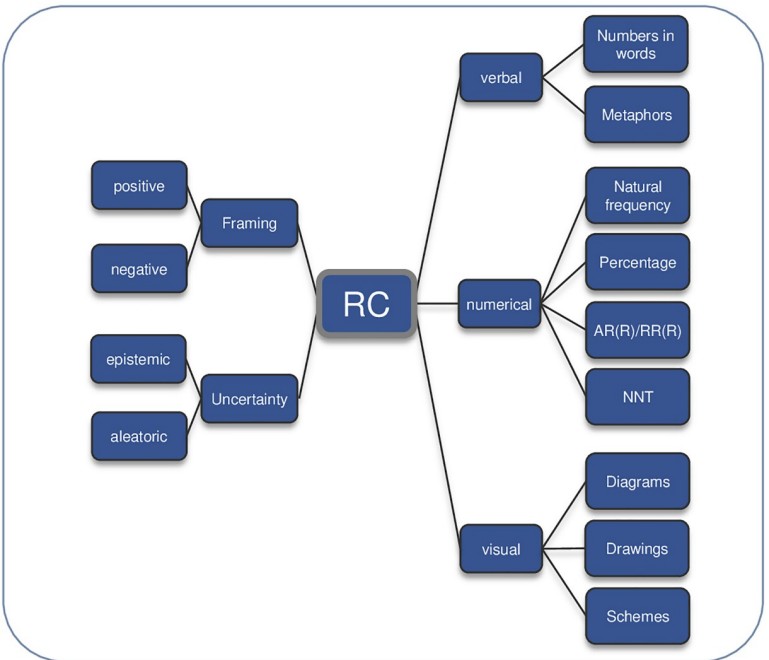

**Fig 1. Framework Risk Communication (RC).** AR(R) = Absolute Risk (Reduction), RR(R) = Relative Risk (Reduction), NNT = Number Needed to Treat.

factors (patient, clinician, consultation) and challenges of risk communication were captured during the data collection process using an inductive approach. Subsequently, we created a new framework with the aim of reflecting the complexity of risk communication in daily clinical practice (Fig 2, Results section).

### Study population and sampling strategy

We used convenience sampling facilitated by the research team's network in Germany and the Netherlands to identify clinicians who were (a) experienced or at least familiar with the concept of SDM, and (b) known to regularly need to communicate risks to their patients. In order to obtain a broad spectrum of perspectives, sampling was based on gender, age, experience and clinical field (general practice, gynecology, nephrology, neurology, genetics, surgery and orthopedics) [51]. We invited 16 clinicians via e-mail, one of whom declined due to lack of time. The final sample consisted of 15 experts. Sampling was part of the iterative process of data collection and analysis and was stopped once data saturation was reached.

### Data collection method

The primary female researcher (RR) carried out semi-structured qualitative interviews [52] at a location preferred by the interviewee. The participants were not known to the primary researcher (RR). Based on the literature review, a semi-structured interview guide (S1 Appendix)

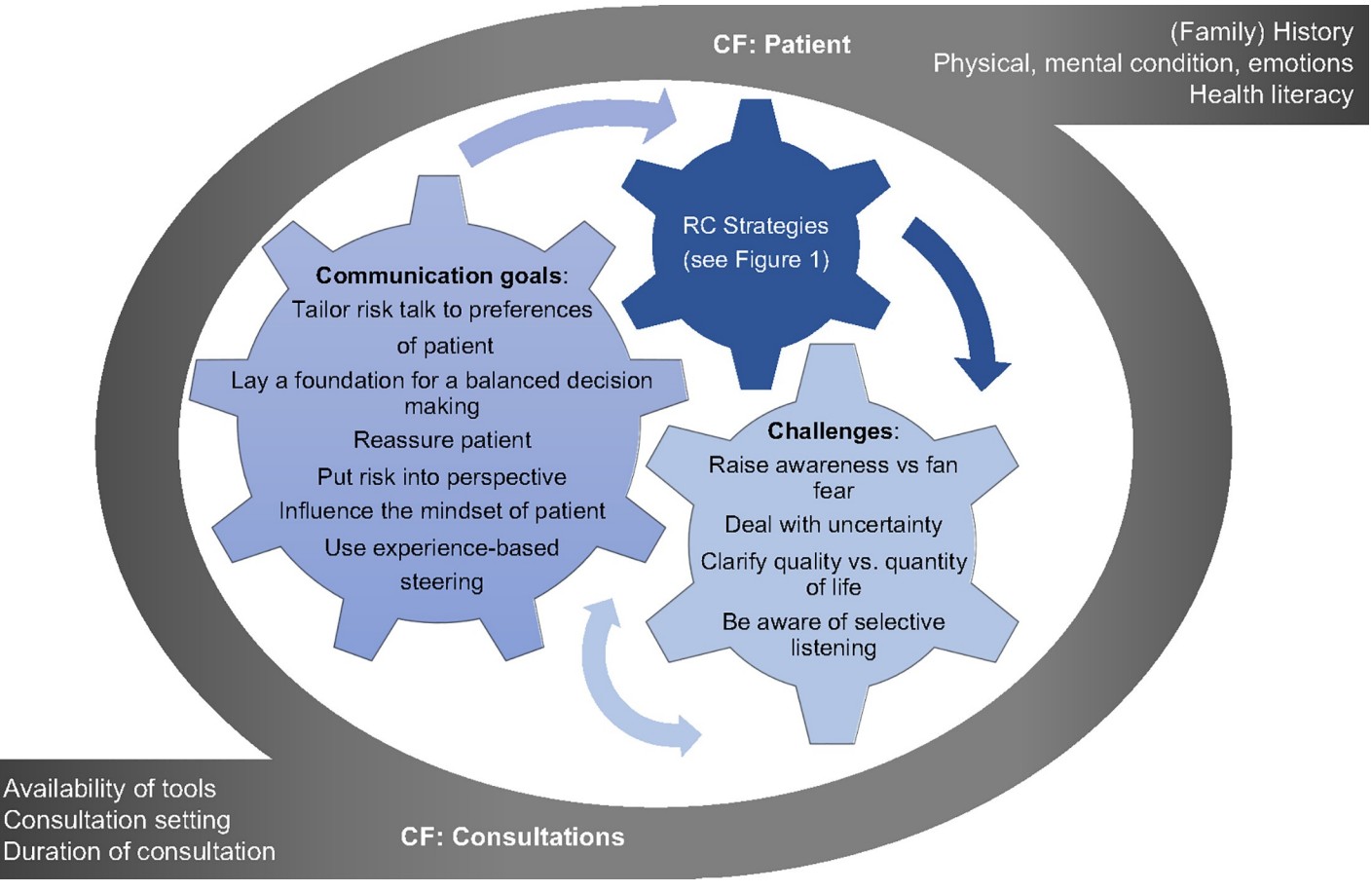

**Fig 2. Framework clinicians' communication goals and challenges in Risk Communication (RC) and influencing Contextual Factors (CF).**

with open-ended questions was developed. In the introduction section of the interview participants were again informed about the goal of the research and consequently asked whether they would like to participate (S1 Appendix). Verbal consent was obtained in the interview session just before the audiotape was started. A pilot interview was conducted to test the preliminary interview guide [52]. Firstly, background information about the professional role was obtained. Interviewees were asked to give their definition of risk communication. Initially interviewees were asked to illustrate their risk communication strategies using pre-existing cardiovascular disease (CVD) or oncology patient cases. After two interviews it became obvious that the best approach to reveal authentic strategies was induced through self-chosen patient cases. Questions were posed according to the five risk communication categories (Fig 1). Insight accrued during the course of data collection led to minor adjustments in the interview guide.

## Data analysis

All interviews were audiotaped and transcribed verbatim. Additionally, drawings made by interviewees were added to the transcript. We started with a deductive content analysis to gather risk communication strategies according to the categories described in the framework. Collaterally, the inductive approach was used to capture and group new aspects concerning clinician goals, challenges and dilemmas, risk communication strategies and content and context-related factors with no preconceived ideas [53]. Text fragments were coded in the software program QUIRKOS. During the data analysis process a schematic table presenting an overview of risk communication strategies with illustrative example sentences (narratives) and influencing contextual factors was created. Risk communication narratives were defined as illustrative example sentences that were used in the consultation room. Two researchers (RR and EG or TW) independently analyzed the transcripts. Their backgrounds are PhD (EG) and MA (RR) in health sciences and PhD in medicine (TW). The researchers reached consensus on the coding through discussion. Detailed information on the analysis can be found in the AUDIT trail (S4 Appendix).

## Techniques to enhance trustworthiness

The data collection was constantly driven by the research question. The theoretical framework based on the literature review was critically verified, reflected upon and adjusted. Diversity and richness of data was enhanced by including participants from a range of clinician backgrounds. A pilot interview verified the relevance of content of the preliminary interview guide and revealed the need for reformulation and adjustment. The semi-structured interview guide comprising broad questions left enough latitude for the interviewee to answer [52]. The iterative process of data collection and analysis allowed the researcher to steadily inform data collection by refining the focus in subsequent interviews. Moreover, re-examining the data in the iterative data analysis led to a continuous meaning making. A member check enabled the participants to comment on the interview transcripts and the first version of the article thus strengthening the credibility of data [54–58]. Transcripts were always independently coded by two researchers (RR and EG or RR and TW) and discussed. Analysis and interpretation were additionally discussed in the research team (investigator triangulation) [57,58].

## Results

### Sample characteristics

The sample consisted of ten female and five male clinicians of varying ages. Twelve of the 15 interviewees worked in the Netherlands and three in Germany. Among the clinicians were two

**Table 1. Characteristics of the interviewees.**

| Clinician | Gender | Age category | Profession discipline | Country | Interview language |
|---|---|---|---|---|---|
| C01 | male | 60–69 | General Practice | NL | English |
| C02 | female | 40–49 | General Practice-Elderly Care | NL | English |
| C03 | female | 40–49 | General Practice | NL | Dutch |
| C04 | male | 50–59 | General Practice | NL | Dutch |
| C05 | female | 40–49 | Gynecology | NL | Dutch |
| C06 | female | 50–59 | Radiotherapy—Oncology | NL | English |
| C07 | female | 40–49 | Internal Medicine—Nephrology | NL | English |
| C08 | male | 30–39 | Internal Medicine—Hematology | NL | Dutch |
| C09 | male | 60–69 | Head and Neck Surgery—Oncology | NL | English |
| C10 | female | 30–39 | Neurology | GE | German |
| C11 | female | 30–39 | Neurology | GE | German |
| C12 | female | 40–49 | Neurology | GE | German |
| C13 | female | 60–69 | Clinical Genetics | NL | English |
| C14 | female | 50–59 | Clinical Genetics | NL | Dutch |
| C15 | male | 30–39 | Orthopedics | NL | Dutch |

nurses, three specialist-in-training and ten physicians. Table 1 gives a detailed description of their characteristics.

## Interviews

The average length of interviews was 34 minutes. Eleven interviews were conducted face-to-face in a relaxed office setting and four were conducted over the telephone. One doctor was on-call during the interview, for this reason the interview was interrupted three times. Obtaining narratives for the applied risk communication strategies in clinical practice was more difficult to achieve in some interviews than others. However, by asking prompting questions it was possible to elicit narratives at every interview. Giving the interviewee latitude to choose their own patient case was especially valuable in assuring elicitation of risk communication strategies that are actually used on a routine basis. After 15 interviews saturation of data was reached as no new strategies emerged according to the categories of the framework (Fig 1). In some interviews we were able to obtain more detailed patient cases, which illustrated the routine of risk communication in an exemplary fashion (S2 Appendix). The strong link between SDM and risk communication became evident as interviewees frequently gave comprehensive statements and illustrations, not only concerning the risk communication format itself, but also for choice or option talk.

## Types of risk

Due to the heterogeneous sample, the interviewees covered a variety of diseases and disorders of varying degrees of severity, duration of treatment and impact. These ranged from a number of types of cancer, cardiovascular risk, chronic diseases such as diabetes and multiple sclerosis, and risk-related to parental-diagnostic and embryology/in-vitro fertilization. These diseases and disorders were perceived as being related to various different types of risk (Table 2) that influence the risk communication process.

## Main findings

The main findings will be presented, comprising the clinician communication goals and challenges they face in risk communication and the risk communication strategies they utilize.

**Table 2. Types of risk.**

| Risk of disease/event |
|---|
| • in a healthy patient with an unknown risk, e.g. CVD, cancer |
| • in an unborn child or embryo |
| Risk of recurrence of disease/event |
| • in a patient already diagnosed with e.g. CVD or cancer |
| Risk of deterioration of symptoms /condition/ quality of life |
| • in a patient with a chronic condition e.g. Chronic kidney disease, cancer |
| Risk of dying and end-of life |
| • in a frail patient or one with a chronic condition e.g. chronic kidney disease, cancer |
| Risk of side effects of treatment |
| • in invasive treatment such as surgery |
| • in patients with acute or chronic disease |
| Risk of treatment or diagnostic burden |
| • e.g. in surgery, drug treatment with high number of doses per day |
| • e.g. in invasive, painful or frightening diagnostic procedure |
| Risk of overtreatment |
| • e.g. resuscitation in cardiac events |
| • e.g. screening for disease in healthy persons |

Contextual factors that shape the risk talk will also be introduced. Fig 2 shows the interaction between those elements with contextual factors (at clinician and patient level) shaping risk communication in daily clinical practice (grey circle). Clinicians apply different risk communication strategies (dark blue gearwheel, Fig 2) that are linked to certain communication goals (medium blue gearwheel, Fig 2). During the process of communicating risk, clinicians face multiple challenges (light blue gearwheel, Fig 2). Hence, the three gearwheels interact in risk communication. The results will be described in an interactive way as the Fig 2 illustrates. Findings will be described and underpinned with illustrative quotes and drawings. The narratives in the quotes are presented in italics. Additional narratives and drawings can be found in S3 Appendix. For ease of reading the keywords used in Fig 2 are highlighted in bold in the results section.

Overall, participants regarded risk communication as an integral part of SDM that allows for balanced decision-making, and to which appropriate time and consideration should be given.

> We are actually moving pretty quickly from risk communication to "*Well, these are your options—and what are we going to do*?*".* So [. . .] going too quickly through the circle of shared decision-making. While [. . .] if people understand this [risk communication] well, I think they are more receptive to the following steps [of shared decision making].
>
> —Interview C04 General Practice—

Thus, in this sample of clinicians, it was not questioned whether or not to conduct a 'risk talk' but rather how to communicate risks when facing each unique patient and situation. The participants emphasized that the complex process of risk communication is highly dependent on a number of contextual factors related to the consultation and patient (grey circle, Fig 2). Clinicians stated that the **consultation setting** whether in a hospital or other type of practice and the **availability of tools** and **time** for the risk talk may shape the risk communication.

They agreed that risk communication is highly dependent on the patient's unique context (such as patient and **family history** and **his/her physical and mental condition**). Clinicians consistently named **health literacy** as an important factor in influencing the risk talk. Overall, interviewees viewed a certain minimum level of health literacy as conditional for effective risk communication. Some clinicians deliberately skipped the risk talk in patients with a low level of literacy. One doctor emphasized the use of a stepwise approach: first estimate the health literacy level of the patient and subsequently adjust the risk talk.

> First, I check for [level of] health literacy. If that is low, I will skip most of my [risk] talks and try to reach them on their level, somewhere. [. . .] for risk communication there needs to be some kind of standard or platform that people are on already.
>
> —Interview C04 General Practice—

During the course of data collection, it became evident that risk communication is clearly shaped by clinicians' own risk communication strategies and their communication goals. The clinicians frequently aimed to **tailor the risk talk to the preferences of the patient** for certain medical options. This approach was reported to have an impact on whether clinicians go into detail when giving information about a certain medical option. One doctor mentioned that tailoring the risk talk to the preferences of the patient reduces the amount of risk information to deliver. That goal was to counteract the challenge of **selective listening by the patient** due to the amount of information delivered.

> So, I put a lot of effort in discussing with the patient what is important for them. What are their goals for the short term and for the long term? And [. . .] when I know what is important [for them], only after that I start discussing risk. [. . .] And then I explain, *"Well I'll tell you about this treatment and that treatment. I haven't told you a lot about this third treatment because what I understand from you is that you don't want an operation"*–an MRI for example. [. . .] And that helps me a lot with my risk communication because then the amount of risks that need to be discussed will decrease.
>
> —Interview C02 General Practice—

By means of the risk talk clinicians seek **to lay a foundation for a balanced decision-making**. Participants reported that they applied different risk communication formats in their consultations, such as verbal and numerical risk communication. Overall, clinicians agreed that communication in verbal terms only leads to ambiguity in definition of the risk size. Nonetheless, some clinicians reported that they describe the risk in verbal terms only in situations where numbers were not available due to lack of knowledge for example, or the non-application of a tool that could have provided information about the size of the risk. Another justification for verbal phrasing of risks was situations in which clinicians passed on information about a very small or a very high risk.

> But sometimes, I must admit that I do use those words occasionally. [. . .] I let myself be tempted to say there is a very small chance [. . .] especially if the odds are very high or very low, then I say so. So, if 95% of people recover, I don't say, 19 of the 20 recover but then I often say, *"Almost everyone recovers."*
>
> —Interview C08 Internal Medicine—Hematology—

Most clinicians stated that they prefer natural frequencies instead of percentages because natural frequencies are easier for patients to understand. A small number of clinicians reported that they only use percentages to communicate a risk estimate. Interviewees said that their choice to use this format was influenced if in their estimation the patient had a higher level of numeracy. Some clinicians reported using both percentages and natural frequencies. Most participants felt that percentages are quite abstract and should be avoided.

> Then I say, "And now there is a side effect that, do not be afraid, it sounds now a bit scary, but it is very rare, however it is very dangerous, and you have to know that it exists. Namely in XX of 1000 people, a [. . .] encephalitis can occur [. . .]."
>
> —Interview C11 Neurology—

If a change in risk was reported, the participants stated that they had a preference for absolute risk since the presentation of relative risk alone can lead to biased perception of risk size as the reference value is missing. Interviewees seldom mentioned the concept of number needed to treat (NNT).

> But the absolute [risk] is much more important than the percentage, I think. With that [relative risk] you can impress a patient: 'You have a 50% risk reduction.' But it depends on how big the baseline risk is and what the absolute risk is. So, I think the relative risk reduction is not so exciting.
>
> —Interview C03 General Practice—

In the process of achieving their goal of laying a balanced foundation for decision-making, clinicians encounter a number of challenges in daily clinical practice. They repeatedly stated that it is difficult for patients to comprehend abstract risk estimates. Some clinicians talked about the influence of the risk size that imposes additional challenges to risk communication, as it impacts on the patient's risk perception. For example, some participants stated that the patient´s emotional arousal may influence their perception of risk, and potentially their interpretation of the size of the risk, therefore the right time point for risk communication should be carefully chosen. Some clinicians reported that the indirect experiences of the patient's significant others seem to have an impact on the patient's perception of risk. Clinicians observed that the occurrence of disease can bias the patient´s sense of risk size towards 100%. They also mentioned that patients apparently do not always relate the population-based risk estimate to themselves or may deny having been informed about a certain risk at an earlier consultation—in the setting of clinical genetics for example—when a small risk of a serious outcome had indeed been communicated by a clinician. Furthermore, in the process of informing the patient, clinicians reported that they are frequently confronted with the challenge of **"raising awareness of a risk versus fanning fear and anxiety in patients".** This aspect seems difficult to balance for the clinicians, as on the one hand they aim to inform patients that real risks do exist and on the other hand, they do not want to frighten the patient.

> You want to do it in a way that is understandable, you want them to understand that there are real risks, but you also don't want to make them too anxious and to scare them, because sometimes you can also induce fright through risk communication.
>
> —Interview C15 Orthopedics—

Hence, the framing of risk messages was regarded as important. Clinicians reported trying to use both positive and negative framing, but only a few clinicians stated that they always applied the two strategies in order to be as neutral as possible. One clinician had a clear preference for positive framing, which the interviewee explained was related to their own personality. Interviewees said that their choice of strategy was influenced by the context, i.e. if the patient was healthy or suffering from a disease, the size of the risk, and patient emotions and patient preferences. Clinicians described using negative framing in high-risk patients such as those with CVD in order to illustrate and emphasize the risk of a heart attack through smoking. Positive framing was also reported as one means of **reassuring patients**.

> For example, recently on the ward someone said, *"Oh I'm scared that next month I will have the disease back"*. I knew that the chance that the lady would have the disease back in the following month was very small. But it wasn't zero. And if I had said to her *"You have a 1 in 10 chance that you will have the disease back next month"*, she would have been worried the whole evening. But I said to her *"Well the chance is 9 in 10 that it will all go well"* and she was reassured. Then I was actually using risk communication [. . .] framed to the patient and to also support her a little bit.
>
> —Interview C08 Internal Medicine—Hematology—

Furthermore, clinicians said that they tried to reassure the patient by **putting the risk into perspective**. In risk talks about small risks some clinicians reported that they related the small risk to experiences with similar cases, or in general to highlight the high chance of a positive outcome. One clinician also used the illustration of the natural fluctuation in organ function to comfort the patient.

> [. . .] people don't understand that 0.0001%. But what kind of chance is that? [. . .] what I literally say is, *"I have to tell this number to you but in my experience in my practice I haven't seen or I've only seen one or two patients who lost their kidney because of a bleeding after a kidney biopsy I would do many many kidney biopsies and it almost always goes good [. . .]."*
>
> —Interview C07 Internal Medicine—Nephrology—

Overall, clinicians considered graphical support to be very helpful for putting risks into perspective, often in relation to time (e.g. 10-year risk, lifetime risk). Clinicians most frequently reported using population pictograms to illustrate the occurrence of a certain outcome or of side effects. Another tool that was repeatedly mentioned by GPs was a color-coded table of cardiovascular disease risk that illustrates the 10-year absolute risks for various high and low risk categories [52]. Another tool interviewees reported to use was bar charts. A smaller number of clinicians also said they used self-made drawings. Two examples are shown here (for further examples see S3 Appendix). Fig 3 shows an illustration by an oncologist of the likelihood of dying over time in which stem cell transplantation treatment is compared with the natural course of leukemia to make the patient aware of the risk of treatment (Fig 3). Another clinician illustrated the peak age of breast cancer in a diagram (Fig 3).

A few clinicians said they used metaphors or visual analogues to put the risk into perspective. As a reason for using metaphors, one clinician mentioned that metaphors support a patient's comprehension because the picture of a certain item allows them to grasp the size of a certain risk better than abstract numbers alone would. One example of putting a risk into perspective was making a comparison with the lottery. To illustrate risk size, another clinician depicted a 50:50 risk by using weighing scales or the odds of having a son or a daughter. To

| Diagram of mortality in leukemia patients in relation to stem cell transplanation with explanatory narrative | Diagram peak age of breast cancer with explanatory narrative |
|---|---|
| 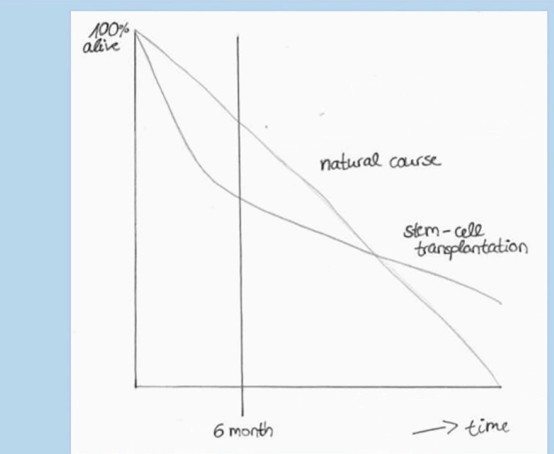 | 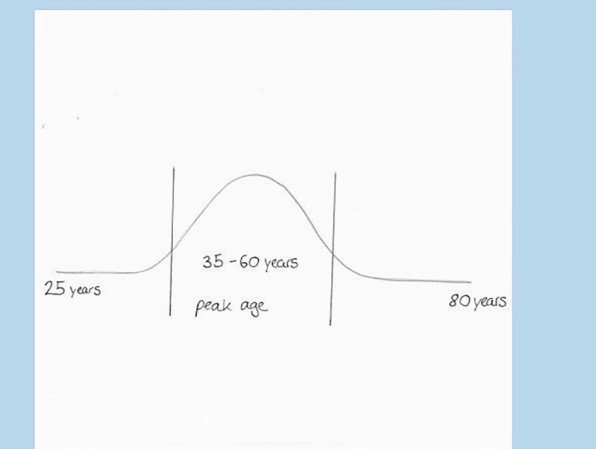 |
| 100% of people are alive on the vertical axis. And [on the y axis] 0% are alive. Then of course everyone starts at the top at 100%. And the natural course of the disease is that people usually die one by one and then in time they all eventually die from that disease. With the stem cell transplantation more people die early on, so you get a steep curve down, but it then bends, and somewhere it crosses the other curve. And that is something that I find difficult to explain verbally;  it is easier to say that you have more risk of dying here in the first six months or so. But ultimately, in the longer term - say two years or so and often two or three years - that you have a greater chance of survival. <br> -- Interview C08 Internal Medicine Hematology -- | So what I do is: make a line like that and then I say here comes a peak age and then it goes down again. Here at the beginning of that line a woman is 25 years old, at the end she is 80 years old. And that peak age - it is between 35 and 60 years. Now I know that people sometimes don't fully understand that well, so then I say afterwards: "that means that most women who are carriers of this abnormality will have breast cancer during this period, at that peak." <br> -- Interview C14 Clinical Genetics -- |

**Fig 3. Self-drawn diagrams of interviewees to visually support risk communication.**

visualize the concept of uncertainty one clinician explained to utilize the picture of a crystal ball. Another reported using the metaphor of an umbrella to visualize the uncertainty that is inevitably part of life and each individual's subjectivity that leads to differences in risk perception and risk seeking.

> If you look up at a cloudy sky, and you ask ten people if they want to take an umbrella or not, then some of them will always take an umbrella, some never will, and some may, depending on how cloudy the sky is. [...] But it is not about whether I would like to take an umbrella, it is about, *"Do you want to take an umbrella?"* [...] one person might not mind walking around carrying an umbrella the whole day for nothing, because she definitely doesn't want to have her new perm spoiled. And the other person thinks 'I'm really not going to walk around all day with that umbrella.'
>
> —Interview C05 Gynecology—

**Uncertainty** is inherent in the whole risk communication process and clinicians repeatedly spoke of challenges in dealing with this complex concept in their daily clinical practice. Some clinicians reported that they communicated epistemic uncertainty in general descriptive verbal terms in situations where there is a lack of evidence.

> Yeah, I communicated it [uncertainty] a little bit more open. I say, "In your case we don't know exactly what is going to work because there isn't a lot of research supporting therapies or treatments for your disease or for your case. So, we need to have trial and error what is working for you."—Interview C02 General Practice—

All clinicians stated that they communicate aleatoric uncertainty verbally. The use of confidence intervals was regarded as too difficult for patients to understand. Two clinicians who are frequently involved in communication on the prognosis of serious disease reported using the strategy of giving a range instead of a precise risk estimate. These interviewees explained that this approach helps to prevent patients holding on to a fixed deadline which has a strong emotional impact as the patient then expects to die at the forecasted time. Therefore, one clinician only spoke of an average life expectancy if directly asked by the patient.

> In this case [concerning average survival] [...] I don´t find it useful for people to know an exact number. And when people ask: 'Can you tell me what the average survival is?' Well then, I have to say approximately six months, but I try not to do that at the beginning,. [...] I find it very unpleasant to give a set time. Because there just isn't one. We only know the average of 100 comparable patients.
>
> —Interview C08 Internal Medicine—Hematology—
>
> I have seen patients with chance of 95% for cure. And they die. And the other way around, patients incurable and they survive. [...] You never know. [...] if you have 1000 patients [...]. 90% will survive or 90% will die. [...] *"But then again that doesn't say anything about you. We don't know."* [...] doctors who say: 'Ok you have [...] three months because if you have 1000 patients with the same disease, one patient will survive one hour, and one will survive 10 years and the mean is three months.' [...] But I never use these kinds of indications. Because I think it's harmful. Because I had patients and then they said: 'Ok, I was with the medical Oncology and he said I have two months to go.' And then [they think] it's the last visit. And then they said goodbye. And this lady [...] she came back in December, still in a good condition. And we were just a bit talking and [she] said yeah: 'The problem is I gave all my stuff for Christmas away. What shall I do?', I say, *"Yeah, you have to ask it back."*[...] And she died next year in in June.
>
> —Interview C09 Head and Neck Surgery—Oncology—

Some clinicians reported that they aimed to **influence the mindset of the patient**. One interviewee who is involved in the treatment of severe disorders tried to make patients aware that death is an undeniable fact of life, and that treatment does not necessarily result in cure. Therefore, that clinician emphasized that it is important to accept death and to make treatment choices that are not based on fear, but based on life choices consistent with the patient's own values.

> [...] as soon as you accept that you will die, then a lot of uncertainty and a lot of anxiety is gone. Because the death is always behind you. So, if you turn around and say, *"Okay, I know I know you are there. So, one day you win."* [...] Because if you are afraid to die and if that' the reason to have the surgery than you take the wrong decision because you will die anyway. Better to get used to the idea that you will die.
>
> —Interview C09 Head and Neck Surgery—Oncology—

The clinicians said they wanted their patients to understand that despite treatment, the risk or risk reduction usually does not approach 0%. In situations related to a medical test such as screening in asymptomatic patients, clinicians might want patients to understand that the test may also lead to additional risks related to overtreatment. Another aspect of risk communication, which clinicians repeatedly mentioned as challenging to communicate to patients, is the treatment burden in relation to the challenge of **"quality versus quantity of life"**. Quality and quantity were regarded as problematic to weigh against one another since concrete numbers for individual patients are lacking. Therefore, clinicians aimed to influence the patients' mindset by making them thoroughly consider the trade-off between quality and quantity of life. Predominantly in situations related to impactful treatment clinicians aimed to draw the patients' attention to the fact that the treatment might have a negative effect that could lead to a decline in quality of life. This awareness potentially supports the laying of the foundation for balanced decision-making for the patient.

> And if you have tried all kinds of different statins then you also try to put it into perspective and look at it this way *"But how much quality of life have you lost due to the side effects you have?"* And that cannot be expressed as a single estimate, as it is related to the experience of the patient. *"And suppose you belong to that group of patients who have been taking it all these years, but actually do not benefit from it, because it's only some of the 100 people who benefit from it, does that still affect your decision to continue or not?"* [. . .]
>
> —Interview C03 General Practice—

In the complex risk talk the experiences and expertise of the clinicians play an important role. Some clinicians reported the goal of **steering the patient towards a certain decision.** This approach is manifested by the clinician emphasizing a certain risk more than other outcomes. It was reported to be used in situations such as the choice for elective surgery where on the basis on his sound medical insights and experiences with similar cases, the doctor deemed a patient as not such a good candidate for surgery. Thus, this goal was not related to steering the patient towards the doctor's treatment preference but to make high risk patients thoroughly consider the side effects of treatment.

> [. . .] in a patient when you explain a certain surgical therapy. But you think that the patient is not entirely eligible for it, or that he or she behaves in a certain way, or has a certain other comorbidity [. . .] then you also use risk communication to emphasize that there really is [a risk]–thus you make those risks a little bit bigger in your explanation.
>
> —Interview C15 Orthopedics—

## Discussion

### Main findings

The aim of this study is to give an overview of clinician's strategies to communicate risk in daily clinical practice with illustrative examples. Risk communication in this study is shaped by various contextual factors at the consultation and patient level, as well as clinicians' specific communication goals. When communicating risk to patients, clinicians face various challenges and uncertainties that may go beyond choosing the appropriate risk format. The majority of risk communication literature seems to focus on examining different risk communication formats that are often isolated from the daily clinical context. [5,10–13,17,22]. Our findings reveal

what actually shapes risk communication in clinical practice, thereby elucidating the interaction between the expertise of clinicians and their specific communication goals, the choice of a certain strategy and the challenges they face in daily practice.

Clinicians apply a number of risk communication strategies in order to lay a foundation for balanced decision-making. Consequently, clinicians make the patient aware of a certain risk. As pointed out by the extensive literature on judgement and decision-making, the risk communication process is prone to heuristics and biases [59–63]. Raised awareness of the risk through a risk talk leads to further challenges for clinicians to address. The patient´s risk perception plays a major role: a small risk of a severe disorder or a medium risk of a mild disorder can be perceived differently. Depending on the context, in the field of embryology for example, the communication of small -but nonetheless impactful- risks can strongly affect the emotional impact on the patient. Thus, clinicians are repeatedly confronted with the challenging dilemma of raising awareness and fanning fear in patients, emphasizing the importance of phrasing the risk message sensitively. There seems to be a need for clinicians to develop skills/strategies to communicate risks without increasing fear [64]. As shown in our study, clinicians repeatedly tried to reassure patients by putting the risk into perspective by using comparisons with experiences from other cases, or the use of visual formats. Some participants stated to use verbal analogues or metaphors to put risks into perspective. Although there is not much to be found in the scientific literature regarding the use of metaphors and visual analogues [65,66], it seems to be a relevant concept, especially in relation to the illustration of aspects of uncertainty. Clinicians noted that the concept of the uncertainty that is inherently present in life is challenging to explain to patients. It is difficult for patients to grasp that numbers are only estimates based on a sample of the population. Furthermore, it is important for patients to be aware of the illusion of certainty, therefore some clinicians tried to influence the mindset of patients by making them aware of aspects of uncertainty in decision-making and life in general: e.g. treatment does not necessarily mean cure and despite the treatment a risk hardly approaches 0%. Therefore, thorough consideration of the aspects of the quality and quantity of life in life-threatening treatment decisions for example, is important to patients. It should be noted that clinicians need to develop a certain expertise concerning the more sensitive aspects of risk communication such as a patient's emotions.

Previous literature on doctor-patient communication empirically supports both the context-specific and goal-directed nature of communication [67–71]. As presented in this study, the risk communication process is influenced by different contextual factors at the consultation, patient and clinician level. Risk communication has to be sensitive to the context of the consultation (context-specific), e.g. concerning practical aspects such as time pressure or patient aspects such as their emotions. Risk communication is also goal-directed as clinicians apply specific communication goals in the process. Those specific communication goals vary and might depend on e.g. patient preferences, but they can also be driven by clinicians' preferences and expertise. For example, clinicians might aim to influence the lifestyle of the patient or to make high risk patients thoroughly consider the side effects of a treatment. Clinicians adapt how they present risk accordingly. Therefore, risk communication is not merely to be satisfied with a standardized stepwise approach of the "right" risk communication format/strategy, considering the uniqueness of situations in daily clinical practice. A holistic, context-specific and goal-directed approach to teaching risk communication seems to be needed.

The complexity of risk communication also became evident during the search for a theory/framework that could support the data collection and analysis process of this study. There was little guidance on a framework for risk communication in the clinical consultation setting. A larger body of scientific literature on risk communication focuses on risk communication concepts in catastrophe or crisis management [72–75]. Research on risk communication in the

clinical setting was rather focused on effectiveness of single risk message formats only, and isolated from the context [13,17–19,21,22].

Overall it can be stated that in concordance with findings and recommendations in the literature, participants used natural frequencies, absolute risk and positive and negative framing in their risk communication. Some clinicians also reported to use percentages. This approach is generally not highly recommended in the scientific literature [6,10,21,22], however recent findings suggest that natural frequencies do not foster better understanding and thus pointing to an equal performance of percentages and natural frequencies [20,76].

In agreement with scientific recommendations, our participants stated to prefer the presentation of absolute risk since relative risk can be misleading in risk perception [12,20]. Concerning visual tools most clinicians were familiar with population pictograms and applied them on a regular basis. Overall, all interviewees agreed that a form of visualization is important to support risk communication, as shown in previous research [28,77–79]. A noteworthy fact is that clinicians did not consistently specify the time frame around the risk estimates. Concerning the communication of numbers in words only, a small number of clinicians reported that they used this approach in some situations, although this is contradictory to the recommendations in the literature [19,20,41]. Clinicians reported that they dealt with uncertainty in descriptive verbal terms. Complex approaches such as confidence intervals were not reported to be used. However, in relation to serious life-threatening illnesses, clinicians gave a range of life expectancy to patients to prevent the negative effects of holding on to a fixed time point.

## Strengths and limitations

As far as we know no other study has given a comprehensive insight into risk communication strategies and narratives that clinicians report to apparently use in practice, in the way that we have done. We provide a comprehensive overview of a sample of clinicians from the Netherlands and Germany. However, as we interviewed a convenience sample of clinicians that are at least familiar with the concept of SDM, these findings might not be generalizable to clinical practice in general, as other doctors might be less inclined to use the presented risk communication strategies. Most of the interviews were conducted in the mother tongue of the participant (Dutch or German) with the exception of six interviews that were conducted in English. Using the mother tongue of the interviewee facilitates the elicitation of authentic narratives. Nevertheless, a drawback might be that the interviewer has only recently learned Dutch. However, discussing the data in the project team with Dutch speaking team members, strengthened the credibility and scientific quality. Another drawback is the small sample size. The findings need to be confirmed in a larger sample size. However, it needs to be considered that qualitative research aims to give a detailed explorative insight rather than confirming specified hypotheses in a representative sample. We are confident that the results reflect adequately the information provided by this sample of clinicians as recruitment and data collection took place until saturation of data was reached. Further, the utilization of interviews to obtain the risk communication strategies relied on the recall of the clinicians. Observations of consultations would probably have resulted in additional information. As observations were not feasible in our study, we believe that using participants' self-selected patient cases sufficiently enhanced authentic reflections about their risk communication process. In addition, this study cannot make any inferences and statements about the effectivity of the illustrated strategies. Nonetheless, these findings could facilitate specific hypotheses that could be tested in future experimental and systematic studies. For reasons of feasibility the patient perspective could not be explored in this study.

## Conclusion

In conclusion, our study gives an insight into what shapes risk communication and what authentic risk communication strategies clinicians from different medical disciplines report to use in daily clinical practice. We illustrated the multi-dimensional nature and complexity of risk communication which is about more than conveying a risk message in an adequate format. The risk communication process is inherently incorporated in the interaction with the unique clinical circumstances. Hence, we conclude that risk communication cannot be taught without taking the context and the clinicians' expertise and communication goals into consideration.

### Implication for further research and practice

Undoubtedly, research on effectiveness of certain risk communication formats is important, but risk communication should be viewed in relation to its context. Further research should address more practical approaches for a risk communication framework in the clinical setting to be able to better examine the topic. The challenge for future research is to take the perspectives of clinicians and patients into account and also the contextual situation related to the disease and type of risk instead of merely focusing on a single format. It could be of interest to elucidate patient satisfaction concerning the described strategies. Further research needs to address the aspect of patient´s health literacy and risk communication. The effectiveness of using illustrative examples of risk communication should be evaluated in a systematic way, as they might be a good method of supporting risk communication training for trainees. Given the important role of context in risk communication, providing clinicians with context-specific risk communication strategies seems important. Hence, the identified strategies can be used as a starting point for the development of medical curricula that address teaching risk communication in a more practical illustrative way, e.g. centered in proactive workplace-learning [44,80], where the expertise of experienced doctors is taken into account. Additionally, the provided risk communication strategies may be of interest to other clinicians.

## Supporting information

**S1 Appendix. Semi-structured interview guide.**
(PDF)

**S2 Appendix. Patient cases that illustrate risk communication.**
(PDF)

**S3 Appendix. Tables risk communication strategies and narratives.**
(PDF)

**S4 Appendix. AUDIT trail.**
(PDF)

**S5 Appendix. COREQ checklist.**
(PDF)

## Acknowledgments

The authors thank Albine Moser for the advice concerning questions on the qualitative methodology, Anke Steckelberg for the contribution to recruitment process and Simona Dumitrescu and Simon Voß for the critical input on the manuscript text and figure.

## Author Contributions

**Conceptualization:** Romy Richter, Esther Giroldi, Jesse Jansen, Trudy van der Weijden.

**Data curation:** Romy Richter, Trudy van der Weijden.

**Formal analysis:** Romy Richter, Esther Giroldi, Trudy van der Weijden.

**Investigation:** Romy Richter.

**Methodology:** Romy Richter, Esther Giroldi, Trudy van der Weijden.

**Supervision:** Trudy van der Weijden.

**Visualization:** Jesse Jansen.

**Writing – original draft:** Romy Richter.

**Writing – review & editing:** Romy Richter, Esther Giroldi, Jesse Jansen, Trudy van der Weijden.

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
