## [Decision Letter · Decision Letter 0]

12 May 2020

PONE-D-20-06323

A qualitative exploration of clinicians´ strategies to communicate risks to patients in the complex reality of clinical practice

PLOS ONE

Dear Dr Richter,

Thank you for submitting your manuscript to PLOS ONE. After careful consideration, we feel that it has merit but does not fully meet PLOS ONE’s publication criteria as it currently stands. Therefore, we invite you to submit a revised version of the manuscript that addresses the points raised during the review process.

We would appreciate receiving your revised manuscript by 11 June 2020. To enhance the reproducibility of your results, we recommend that if applicable you deposit your laboratory protocols in protocols.io, where a protocol can be assigned its own identifier (DOI) such that it can be cited independently in the future. For instructions see: http://journals.plos.org/plosone/s/submission-guidelines#loc-laboratory-protocols

We look forward to receiving your revised manuscript.

Kind regards,

Andrew Soundy

Academic Editor

PLOS ONE

Journal Requirements:

2. Please provide additional details regarding participant consent. In the ethics statement in the Methods and online submission information, please ensure that you have specified how verbal consent was documented and witnessed.

Additional Editor Comments (if provided):

Mention and make sure you adhere to a qualitative framework and mention which one e.g., O’Brien et al., 2014 or Tong et al., 2007 and include in a supplementary file

Give detail in supplementary file of any changes to semi structured interview after piloting

Give examples of the analysis within an audit trail - so examples of each stage so the reader could take the data and follow through - or replicate the stages - no need for full interview manuscripts just examples that show clearly the steps

Identify the ontological position of your selected methodology too please

Make sure the issues of sample size are considered and fit with your pragmatic position

Check if member checking and triangulation fits with your paradigmatic position and please reference this for the reader so I can check this.

Reviewers' comments:

Reviewer's Responses to Questions

**Comments to the Author**

1. Is the manuscript technically sound, and do the data support the conclusions?

Reviewer #1: Yes

2. Has the statistical analysis been performed appropriately and rigorously? 

Reviewer #1: N/A

3. Have the authors made all data underlying the findings in their manuscript fully available?

Reviewer #1: Yes

4. Is the manuscript presented in an intelligible fashion and written in standard English?

Reviewer #1: Yes

5. Review Comments to the Author

Reviewer #1: Thank you for the opportunity to review this article, which examines experienced clinicians’ strategies for communicating risk in the context of their clinical practice. I read the article with interest: the authors have managed to present the at times complex literature on risk communication in a transparent way (Figure 1) and adopt a design that successfully bridges theory and clinical practice. The findings point to the importance of contextual factors in risk communication and relatedly the discursive context of shared decision-making conversations, factors which the authors point out are missing from existing risk communication guidelines and literature. The article is very well-written throughout, with a clarity that stems from the writing but also from the supporting figures and well-selected quotes. As the authors argue, the findings have practical implications for advanced communication skills for junior doctors/ trainees. My comments are mainly to do with points of clarification and suggestions, including phrasing suggestions, which are outlined below.

Abstract

- Suggest there is a word order issue: ‘but fan fear of anxiety’ should be ‘but fear fanning anxiety’.

Background

- Lines 76-77: the choice of ‘manipulation’ is quite strong and implies a negative intention on the behalf of the clinician to my reading. The data (quotes) however from the participants suggest the choice of framing is considered and patient oriented. Alternatively, ‘Choice of framing can influence’ etc.

- Communicating risk to patients in daily clinical practice (lines 93-101, p4): most of the references in this section are from general practice, so it was unclear to me whether ‘daily clinical practice’ referred to primary care or was intended to be more generic. However, a major finding was the importance of clinical context (the disease setting as well as patient characteristics), yet this isn’t foreshadowed in this introductory section. Perhaps continue to build the research gap after line 97 by giving examples of the likely risk communication settings/ challenges for different clinical contexts, in which the ‘daily clinical practice’ will look very different.

- Lines 103 104 p.4, read like the first line of the conclusion. Instead – the aim is to …

- I suggest the phrase ‘young doctors’ is replaced by junior doctors or simply ‘trainees.’ They are not always ‘young’ by the time they finish their training!

- The introduction is very well written, providing a strong rationale for the research.

Methods

The study design is thoughtful, fit for purpose and creative. Figure 1 is likely to be useful for educators (as are the findings).

I also like the transparency of research practices e.g. in lines 155-156, p6

Data analysis: I would like some clarification on the inductive analysis. Lines 163, page 6 mentions themes but the reference, 50, is to qualitative content analysis rather than thematic analysis. I suggest the analysis was content analysis.

The section on techniques to enhance trustworthiness is exemplary because it covers a range of processes in the methods, not just analysis. For the analysis, however, the data analysis sentence p. 177 refers to going back to the interviews. There is little on how the transcripts were coded for the inductive component -here and in the preceding data analysis section. This could be elaborated.

Results

Table 2 is illuminating about the types of risks communication and the contexts in which they occur. Figure 2 is also instructive, and I can see the applications of these tables and figures to teaching. The quotes are well chosen to support the construct presented.

Pl0, line 238. Should it read So […] to too quickly going through.. evaluative too, or to? Both work but different meanings.

The finding in line 260 p. 11 is important and is supported by the quotes; I don’t think this findings is sufficiently reflected in the abstract ie noted the ‘goal oriented’ nature of the clinician’s communication is really part of the SDM (informed by the patient’s goal) and not a paternalistic, clinician centred goal.

Discussion

P. 19 for clarification: in the recommendation for practice “therefore, an holistic, context-specific and goal-directed approach…” do the authors mean that the approach to risk communication should be sensitive to context rather than context specific, where context specific refers to teaching risk communication approaches tailored to a particular clinical discipline? I think the message is the former rather than the latter as it is the principle. It might also be worth clarifying here and in the conclusion (line 546) that the clinician’s communication goals are informed by the goals of the patient.

6. PLOS authors have the option to publish the peer review history of their article (what does this mean?). If published, this will include your full peer review and any attached files.

Reviewer #1: Yes: Robyn Woodward-Kron

---

## [Author Response · Author response to Decision Letter 0]

11 Jun 2020

Editor comments

1. Please ensure that your manuscript meets PLOS ONE´s style requirements, including those for file naming.

I adjusted the following things:

• Changed name “Manuscript-RC” to “Manuscript”

• Adjusted the headings so that only the first word is capitalized

• Table text has been formatted in black

• Supplementary information has been mentioned after references

• Separate heading for authors contribution in manuscript text has been deleted

2. Please provide additional details regarding participant consent. In the ethics statement in the Methods and online submission information, please ensure that you have specified how verbal consent was documented and witnessed: 

We explained the procedure of informed consent more in detail. Under ethical approval and informed consent p.5, line 131-136 and data collection method p.6, line 167-169.

p.5, line 131-136: The NVMO Ethical Review Board granted approval for our study. Clinicians were informed about the study via e-mail. In this e-mail the research was described, and the goal was stated, and the clinicians were asked whether they like to participate in a qualitative interview. Subsequently, clinicians gave written informed consent (via e-mail) whether they were willing to participate. Further, verbal consent was obtained before the start of the interview, just before the audiotape had been started. Verbal informed consent was given for audiotaping and using the data for publication in a scientific journal. Verbatim transcripts of the recorded interviews were anonymized with codes.

p.6, line 167-169: […] In the introduction section of the interview participants were again informed about the goal of the research and consequently asked whether they would like to participate (S1 Appendix). Verbal consent was obtained in the interview session just before the audiotape was started. 

3. Mention and make sure you adhere to a qualitative framework and mention which one e.g., O’Brien et al., 2014 or Tong et al., 2007 and include in a supplementary file:

We clarified this aspect by mentioning the qualitative framework in the manuscript text p.5, line 128-129. We attached the filled-in COREQ checklist as the S5 Appendix. p.5, line 128-129: […] The consolidated criteria for reporting qualitative research (COREQ) has been used to guide the reporting of the research (49). 

4. Identify the ontological position of your selected methodology too please: 

We have added the information in a sentence that explicitly mentions the ontological orientation p.5, line 124-125. p.5, line 124-125: We conducted a qualitative descriptive study with semi-structured interviews. We took the ontological position of relativism, which holds the view that multiple subjective realties exist. Consequently, we chose the epistemological orientation of subjectivism referring to […]

5. Give detail in supplementary file of any changes to semi structured interview after piloting: In the Interview Guide (S1 Appendix) comments were added that indicate the changes made after piloting. The following comments were added:

•After the first interview we adjusted some numbers in the CVD patient case 

•After the first interviews it became obvious that the patient cases, were not necessary to make the participants talk about their risk communication experiences. Moreover, most interviewees spontaneous reflected on their own cases with regard to this topic. However, the patient cases were left in the interview guide in case of unexpected need but were not actively integrated in the interview 

•After the first two interviews the interviewee used the following structure concerning the RC methods: After letting the interview freely talk about his risk communication strategies, the interviewee paraphrased the content and if a certain risk communication category (see figure 1) was missing the interviewer asked explicitly for it. However, phrasing of questions were broad enough to leave the interviewee latitude to answer. 

•After the first two interviews the terms aleatoric and epistemic uncertainty were actively distinct and used when asking about uncertainty

6. Give examples of the analysis within an audit trail - so examples of each stage so the reader could take the data and follow through - or replicate the stages - no need for full interview manuscripts just examples that show clearly the steps: The AUDIT Trial has been added as a supplementary file (S4 Appendix). 

7. Make sure the issues of sample size are considered and fit with your pragmatic position: 

We carefully considered your comment on the sample size. We acknowledge that small sample sizes have limitations. However, we think the sample size fits with our pragmatic position. It is rather a small sample, but we collected data until data saturation was reached and no new aspects emerged. We are aware that the findings of this small sample give only an explorative insight into the clinician’s experiences and the strategies would need to be confirmed in a larger sample. We have elaborated on this issue and expanded the limitations of the sampling method and sample size in the discussion section p.19, line 555-560

p.19, l. 555-560: […] Another drawback is the small sample size. The findings need to be confirmed in a larger sample size. However, it needs to be considered that qualitative research aims to give a detailed explorative insight rather than confirming specified hypotheses in a representative sample. We are confident that the results reflect adequately the information provided by this sample of clinicians as recruitment and data collection took place until saturation of data was reached. 

8. Check if member checking and triangulation fits with your paradigmatic position and please reference this for the reader so I can check this: 

Since we took the position that multiple subjective realties exist it leads to the fact that also many interpretations of reality exist (relativism, subjectivism). Therefore, it is of value to conduct a member check as it allows the person who provided the information to check whether the researcher has accurately reported and reflected this information. For this reason, we have sent the Interview transcripts to the interviewees and the first version of the article for feedback. These actions strengthen the trustworthiness of findings and may augment the findings since the interviewee can comment on the accuracy of the transcript and reporting of results (Bradshaw 2017, Frambach 2013, Koelsch 2013, Korstjens & Moser 2018, Lincoln & Guba 1985). 

Triangulation can concern different aspects of the research process. In our case we refer to “investigator triangulation” which points to the fact that at least two researchers of the research team were involved in the organizational aspects of the study and the data analysis process. Interviews were independently coded by two researchers (RR and TW or RR and EG) and subsequently discussed. The researcher RR and TW held regular meetings in the whole process of examination of the study and RR and EG especially in the process of data analysis. Further, the entire research team had regular meetings during the analysis and reporting phase of this work (RR, EG, JJ, TW). The multidisciplinary character of the research team with different backgrounds in health sciences, medicine and psychology enriched the data analysis process.

We have rewritten this part of the paragraph on p. 7, line 198-202 to clarify the messages we attempt to communicate. 

p.7, line 198-202: […] A member check enabled the participants to comment on the interview transcripts and the first version of the article thus strengthening the credibility of data (Bradshaw 2017, Frambach 2013, Koelsch 2013, Korstjens & Moser 2018, Lincoln & Guba 1985). Transcripts were always independently coded by two researchers (RR and EG or RR and TW) and discussed. Analysis and interpretation were additionally discussed in the research team (investigator triangulation) (Frambach 2013, Korstjens & Moser 2018). 

References 

•Bradshaw, C., Atkinson, S., & Doody, O. (2017). Employing a Qualitative Description Approach in Health Care Research. Global qualitative nursing research, 4, 2333393617742282. https://doi.org/10.1177/2333393617742282

•Frambach, J. M., van der Vleuten, C. P., & Durning, S. J. (2013). AM last page. Quality criteria in qualitative and quantitative research. Academic medicine: journal of the Association of American Medical Colleges, 88(4), 552. https://doi.org/10.1097/ACM.0b013e31828abf7f

•Koelsch, L. E. (2013). Reconceptualizing the Member Check Interview. International Journal of Qualitative Methods, 168–179. https://doi.org/10.1177/160940691301200105

•Korstjens, I., & Moser, A. (2018). Series: Practical guidance to qualitative research. Part 4: Trustworthiness and publishing. The European journal of general practice, 24(1), 120–124. https://doi.org/10.1080/13814788.2017.1375092

•Lincoln, Y. S., & Guba, E. A. (1985). Naturalistic inquiry. Beverly Hills, CA: Sage. 

Reviewer comments

Abstract

- Suggest there is a word order issue: ‘but fan fear of anxiety’ should be ‘but fear fanning anxiety’.: 

Thank you for your suggestion. In the abstract p.2, line 40 it says fan fear or anxiety in patients. We referred to fear and anxiety since it does not necessarily always result in fanning fear but possibly a milder form, namely anxiety. In order to prevent confusion, we decided to rephrase the sentence. 

p.2, line 40: Clinicians try to lay a foundation for balanced decision-making and to incorporate patient preferences while faced with several challenges such as the dilemma of raising awareness but triggering anxiety or fan fear in patients. 

Background

- Lines 76-77: the choice of ‘manipulation’ is quite strong and implies a negative intention on the behalf of the clinician to my reading. The data (quotes) however from the participants suggest the choice of framing is considered and patient oriented. Alternatively, ‘Choice of framing can influence’ etc.: 

Thank you for your suggestion. We have exchanged “manipulation by framing influences...” for “choice of framing can influence...”. p.3, line 75: Choice of framing can influence the perception of risks and hence the decision made during a clinical consultation (15). 

- Communicating risk to patients in daily clinical practice (lines 93-101, p4): most of the references in this section are from general practice, so it was unclear to me whether ‘daily clinical practice’ referred to primary care or was intended to be more generic. However, a major finding was the importance of clinical context (the disease setting as well as patient characteristics), yet this isn’t foreshadowed in this introductory section. Perhaps continue to build the research gap after line 97 by giving examples of the likely risk communication settings/ challenges for different clinical contexts, in which the ‘daily clinical practice’ will look very different: 

Thank you for raising this point. ‘Daily clinical practice’ referred to primary and secondary health care. We have enriched the text to reflect more on the importance of the clinical context for our research question in different medical disciplines, p.4, line 91-104. 

p.4, line 91-104: […] risk communication remains a major challenge (9,12,21,23). The various medical disciplines with their different contexts demand clinicians to nuance the risk talk sensitively to specific aspects of their discipline. For example, in clinical genetics risk communication needs to address predictive testing such as the small chance of getting a false positive test result or the small chance of miscarriage in invasive testing (43). Thus clinical geneticists often have to deal with communicating small and difficult to imagine risks that could have a great impact in the future of the patient. General practitioners often counsel the patient for more general problems in the present, while doctor and patient usually have a closer and continuous relationship with repetitive opportunities for a dialogue and reassurance of the patient. Whereas surgeons often have to communicate the side effects and consequences of severe surgical procedures under time pressure while having seen the patient one or two times in the hospital. On the other hand, oncologists often have to consider the communication of overtreatment and overdiagnosis when discussing screening and treatment options, especially when dealing with frail elderly patients. Undoubtedly, risk communication is a core skill for clinical counsellors in various medical disciplines. However, as far as we know, the literature mainly provides […] 

- Lines 103 104 p.4, read like the first line of the conclusion. Instead – the aim is to ...: 

Thank you for your suggestion. We agree with you and have adjusted the sentence p.4, line 112-113: The aim is to add to the existing literature on risk communication by gaining a deeper understanding of the clinician’s actual strategies in communicating risk in daily clinical practice. 

- I suggest the phrase ‘young doctors’ is replaced by junior doctors or simply ‘trainees.’ They are not always ‘young’ by the time they finish their training!:

We agree with you and have exchanged the term “young doctors” for “trainees” in the whole manuscript. 

Methods

- Data analysis: I would like some clarification on the inductive analysis. Lines 163, page 6 mentions themes but the reference, 50, is to qualitative content analysis rather than thematic analysis. I suggest the analysis was content analysis.

We indeed used a content analysis. The term “themes” was used as a general term and did not refer to the thematic analysis. However, we agree that using the term “themes” in relation with content analysis leads to confusion. In order to prevent this, the word “themes” has been exchanged through the word “aspects“, p.6, line 181: […] Collaterally, the inductive approach was used to capture and group new aspects concerning clinician goals, challenges and dilemmas, risk communication strategies and content and context-related factors with no preconceived ideas (53). 

- The section on techniques to enhance trustworthiness is exemplary because it covers a range of processes in the methods, not just analysis. For the analysis, however, the data analysis sentence p. 177 refers to going back to the interviews. There is little on how the transcripts were coded for the inductive component -here and in the preceding data analysis section. This could be elaborated.

To clarify the process of data analysis we have added a supplementary file (S4 Appendix) with an AUDIT Trail to explain the analysis process in more detail. 

Results

- Pl0, line 238. Should it read So [...] to too quickly going through.. evaluative too, or to? Both work but different meanings:

We double-checked the quote. The main message is “going too fast through the circle of SDM”, in order to make it clearer we adjusted the word order of the quote, p.10, line 257-261.

p.10, line 258: We are actually moving pretty quickly from risk communication to “Well, these are your options - and what are we going to do?”. So [...] going too quickly through the circle of shared decision-making. While [...] if people understand this [risk communication] well, I think they are more receptive to the following steps [of shared decision making].

-- Interview C04 General Practice -- 

- The finding in line 260 p. 11 is important and is supported by the quotes; I don’t think this findings is sufficiently reflected in the abstract ie noted the ‘goal oriented’ nature of the clinician’s communication is really part of the SDM (informed by the patient’s goal) and not a paternalistic, clinician centred goal: 

You have raised a core aspect of our findings that apparently needs more clarification. Our findings show that next to the risk communication strategies, clinicians also use specific communication goals. The goal- directed nature of risk communication refers to those specific communication goals that are applied in the risk communication process next to the risk communication strategies. One specific communication goal is indeed “patient-driven” as the clinicians often try to tailor the risk talk to the preferences of the patient. However, those specific communication goals do not necessarily have to be “patient preference-driven”. Some goals also depend on the clinician’s preferences and expertise e.g. “use of experienced based steering through the clinician” can be applied to influence the lifestyle of the patient or to make high risk patients thoroughly consider the side effects of a treatment. We have revised the paragraph about the goal-directed and context- specific nature in the discussion. Please also see the next row of this table.

In the abstract we added the goal “tailor the risk talk to the preferences of the patient” in the sentence p.2, line 38-40.p.2, line 39: […] Clinicians try to lay a foundation for balanced decision-making and to incorporate patient preferences while faced with several challenges such as the dilemma of raising awareness but triggering anxiety or fan fear in patients. 

Discussion

- P. 19 for clarification: in the recommendation for practice “therefore, an holistic, context-specific and goal-directed approach...” do the authors mean that the approach to risk communication should be sensitive to context rather than context specific, where context specific refers to teaching risk communication approaches tailored to a particular clinical discipline? I think the message is the former rather than the latter as it is the principle. It might also be worth clarifying here and in the conclusion (line 546) that the clinician’s communication goals are informed by the goals of the patient: 

Thank you for your input. As you already assumed, “context-specific” refers to the fact that risk communication should be sensitive to the context. In order to clarify this matter of phrasing, we added an explanation of our definition of context-specific, p.18, line 509- 517.p.18, line 509-517: Previous literature on doctor-patient communication empirically supports both the context-specific and goal-directed nature of communication (67-71). As presented in this study, the risk communication process is influenced by different contextual factors at the consultation, patient and clinician level. Risk communication has to be sensitive to the context of the consultation (context-specific), e.g. concerning practical aspects such as time pressure or patient aspects such as their emotions. Risk communication is also goal-directed as clinicians apply specific communication goals in the process. Those specific communication goals vary and might depend on e.g. patient preferences, but they can also be driven by clinicians’ preferences and expertise. For example, clinicians might aim to influence the lifestyle of the patient or to make high risk patients thoroughly consider the side effects of a treatment. Clinicians adapt how they present risk accordingly. Therefore, risk communication is not merely to be satisfied with a standardized stepwise approach of the “right” risk communication format/strategy, considering the uniqueness of situations in daily clinical practice. A holistic, context-specific and goal-directed approach to teaching risk communication seems to be needed.

---

## [Decision Letter · Decision Letter 1]

14 Jul 2020

A qualitative exploration of clinicians´ strategies to communicate risks to patients in the complex reality of clinical practice

PONE-D-20-06323R1

Dear Dr. Richter,

We’re pleased to inform you that your manuscript has been judged scientifically suitable for publication and will be formally accepted for publication once it meets all outstanding technical requirements.

Kind regards,

Andrew Soundy

Academic Editor

PLOS ONE

Additional Editor Comments (optional):

Reviewers' comments:

Reviewer's Responses to Questions

**Comments to the Author**

1. If the authors have adequately addressed your comments raised in a previous round of review and you feel that this manuscript is now acceptable for publication, you may indicate that here to bypass the “Comments to the Author” section, enter your conflict of interest statement in the “Confidential to Editor” section, and submit your "Accept" recommendation.

Reviewer #1: All comments have been addressed

2. Is the manuscript technically sound, and do the data support the conclusions?

Reviewer #1: Yes

3. Has the statistical analysis been performed appropriately and rigorously? 

Reviewer #1: N/A

4. Have the authors made all data underlying the findings in their manuscript fully available?

Reviewer #1: Yes

5. Is the manuscript presented in an intelligible fashion and written in standard English?

Reviewer #1: Yes

6. Review Comments to the Author

Reviewer #1: (No Response)

7. PLOS authors have the option to publish the peer review history of their article (what does this mean?). If published, this will include your full peer review and any attached files.

Reviewer #1: **Yes: **Robyn Woodward-Kron

---

## [Editor Report · Acceptance letter]

23 Jul 2020

PONE-D-20-06323R1 

A qualitative exploration of clinicians’ strategies to communicate risks to patients in the complex reality of clinical practice 

Dear Dr. Richter:

I'm pleased to inform you that your manuscript has been deemed suitable for publication in PLOS ONE. Congratulations! Your manuscript is now with our production department. 

Kind regards, 

on behalf of

Dr. Andrew Soundy 

Academic Editor

PLOS ONE